# MaskOCR: Scene Text Recognition with Masked Vision-Language Pre-training

**Pengyuan Lyu, Chengquan Zhang, ShanShan Liu, Meina Qiao, Yangliu Xu, Liang Wu, Kun Yao, Junyu Han, Errui Ding, Jingdong Wang**
*VIS, Baidu Inc.*
*lvpyuan@gmail.com*
*{zhangchengquan, liushanshan07, qiaomeina, xuyangliu, wuliang11 yaokun01, hanjunyu, dingerrui}@baidu.com*
*wangjingdong@outlook.com*

**Reviewed on OpenReview:** *https://openreview.net/forum?id=KNAWoKKpi3*

## Abstract

Text images contain both visual and linguistic information. However, existing pre-training techniques for text recognition mainly focus on either visual representation learning or linguistic knowledge learning. In this paper, we propose a novel approach MaskOCR to unify vision and language pre-training in the classical encoder-decoder recognition framework. We adopt the masked image modeling approach to pre-train the feature encoder using a large set of unlabeled real text images, which allows us to learn strong visual representations. In contrast to introducing linguistic knowledge with an additional language model, we directly pre-train the sequence decoder. Specifically, we transform text data into synthesized text images to unify the data modalities of vision and language, and enhance the language modeling capability of the sequence decoder using a proposed masked image-language modeling scheme. Significantly, the encoder is frozen during the pre-training phase of the sequence decoder. Experimental results demonstrate that our proposed method achieves superior performance on benchmark datasets, including Chinese and English text images.

## 1 Introduction

Text recognition, which involves transcribing visual information into text, is a crucial technology for bridging vision and language. It has a wide range of applications, including visual search, document digitization, and more. Many significant progres has been made in the field of text recognition. However, it remains a challenging task due to the difficulty of recognizing fine-grained categories of text, which can vary in fonts, colors, and other factors, coupled with the relative scarcity of labeled real-world data. As an alternative approach, synthetic data has been used in previous studies (e.g., Jaderberg et al. (2014a); Max Jaderberg and Karen Simonyan and Andrea Vedaldi and Andrew Zisserman (2016); Shi et al. (2017a; 2019); Yu et al. (2020); Fang et al. (2021)), and meaningful results have been obtained. Nonetheless, recognition performance is still restricted by the domain gap between synthetic and real-world data.

Efforts have been made to reduce the need for real labeled data through pre-training, which can be broadly divided into two categories: strengthening visual representation learning with unlabeled real images, and introducing language priors with a language model. In previous studies (e.g., Liu et al. (2022); Yang et al. (2022)), contrastive learning and masked image modeling technologies were employed for pre-training using a large amount of unlabeled image data to obtain better visual representations. In other studies (e.g., Qiao et al. (2020); Fang et al. (2021)), a pre-trained language model was used to guide recognition model learning or correct recognition model predictions. These methods have achieved promising results, thanks to their incorporation of vision or language priors. However, there are two limitations to these approaches. First, they tend to focus solely on either visual representation learning or linguistic knowledge learning, while text images inherently contain both visual and linguistic information. Neglecting either type of prior knowledge may

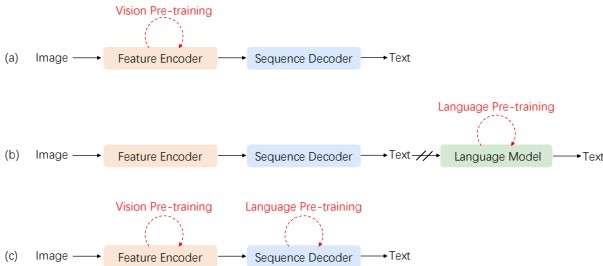

Figure 1: Comparison of previous pre-training methods. A classical recognition model consists of a feature encoder and a sequence decoder, responsible for extracting visual representations and decoding text sequences from those representations, respectively. Previous pre-training methods have primarily focused on pre-training the feature encoder for better visual representation learning (a) or equipping a separate language model to introduce language priors (b). In this paper, we take a different approach by simultaneously and naturally integrating vision and language pre-training into the encoder-decoder recognition model (c).

result in loss of accuracy. Second, previous studies (e.g., Qiao et al. (2020); Fang et al. (2021)) introduced language priors into the recognition model with a detached language model that blocked gradient flow between the recognition and language models, potentially leading to suboptimal performance.

In this paper, we explore the utilization of both visual and language priors through pre-training to enhance text recognition performance. Our approach unifies vision and language pre-training within the classical encoder-decoder recognition framework. Specifically, we pre-train the feature encoder on a large set of unlabeled real text images using masked image modeling, which enables the extraction of better visual representations through self-supervised mechanisms. Additionally, we directly pre-train the sequence decoder to improve language modeling capabilities. To achieve this, we first transform the text corpus into synthesized text images to unify the data modalities and then use a proposed masked image-language modeling technology to learn the linguistic representation. During language pre-training, we fix the pre-trained feature encoder and only update the sequence decoder. This strategy benefits from the language pre-training task, which explores language rules while the pre-trained encoder is not affected by the synthesized text image.

We validate the effectiveness of our proposed approach on Chinese and English text images through extensive experiments and detailed discussion. Our proposed method achieves state-of-the-art performance and significantly surpasses previous methods, particularly on Chinese benchmarks.

The main contributions of this paper can be summarized as follows:

- We propose a masked vision-language pre-training method that unifies vision and language representation learning within the classical encoder-decoder recognition framework.

- Our vision-language pre-training approach contributes to a significant improvement in text recognition. Specifically, with the proposed pre-training technology, we observe a 5% performance gain on the challenging Chinese benchmark BCTR. Furthermore, compared to previous state-of-the-art methods, we achieve superior results on both Chinese and English datasets.

## 2   Related Work

**Text recognition.**  Text recognition can be achieved through various approaches, including character-based Wang et al. (2011); Jaderberg et al. (2014b); Lyu et al. (2018); Liao et al. (2019), word-based Max Jaderberg and Karen Simonyan and Andrea Vedaldi and Andrew Zisserman (2016), and sequence-based Shi et al. (2017a; 2019); Yu et al. (2020); Fang et al. (2021); Xue et al. (2021) methods. Character-based methods recognize text images by performing character localization, classification, and grouping. Word-based methods treat each word as a category and recognize text through image classification. Sequence-based methods treat text recognition as a sequence labeling problem. This approach uses methods such as CTC Graves et al. (2006) and attention mechanism Shi et al. (2019); Lyu et al. (2019); Yu et al. (2020);

Fang et al. (2021) to align the input image patch sequence with the output character sequence. Recently, the sequence-based solution has been the most widely studied framework due to its flexibility and ease of ground-truth labeling. The sequence-based architecture consists of two main modules: feature encoder and sequence decoder. The feature encoder learns visual representations for text images, while the sequence decoder maps the representations into character sequences, with or without the aid of a language module.

**Pre-training.** Representation pre-training has been shown to be beneficial for improving downstream tasks, such as supervised or self-supervised pre-training on ImageNet for computer vision, and self-supervised pre-training for natural language processing. Self-supervised pre-training, which does not require labeled data, has gained significant attention in recent years. Several approaches have been proposed to achieve self-supervised pre-training, such as siamese networks trained using contrastive learning in He et al. (2020); Chen et al. (2020; 2021c), and masked autoencoders for both vision and NLP models in He et al. (2022); Bao et al. (2022); Devlin et al. (2019), where the models improve their representations by predicting masked content. In the method proposed by Rust et al. (2023), pre-training is conducted on text images using Masked Image Modeling to enhance the feature extraction capabilities of the visual encoder, which is similar to the first stage of our method. However, a key divergence from Rust et al. (2023) lies in our method's incorporation of Masked Image-Language Modeling for additional pre-training of the decoder. This addition aims to enhance the semantic reasoning ability of the decoder when interpreting character sequences.

In recent years, there has been significant interest in the field of vision-language pre-training, whereby models are trained to jointly understand both visual and textual information. Various approaches have been proposed for aligning visual and language representations, including image-text contrastive learning Radford et al. (2021); Jia et al. (2021), image-text matching Li et al. (2021a); Wang et al. (2021b), masked language modeling Zhou et al. (2020); Wang et al. (2021a), and language modeling Li et al. (2022); Yu et al. (2022). These pre-trained models can then be fine-tuned for specific downstream tasks, such as visual question answering, image captioning, image retrieval, and text-to-image generation.

**Pre-training for text recognition.** Several methods have been studied to employ pre-training for text recognition. In Jaderberg et al. (2014a); Max Jaderberg and Karen Simonyan and Andrea Vedaldi and Andrew Zisserman (2016); Shi et al. (2017a; 2019); Yu et al. (2020); Fang et al. (2021); Li et al. (2021b), the recognition model is pre-trained on synthesized data using supervised learning. However, despite this approach, the recognition performance is still limited by the domain gap between synthesized and real data. To improve the recognition accuracy, some methods use unlabeled real images or text corpus. For instance, Liu et al. (2022) introduces self-supervised contrastive pre-training to learn representations from the input real images, while Yang et al. (2022) integrates contrastive learning and masked image modeling to pre-train the recognition model. Notably, the aforementioned methods are mainly focused on visual representation learning. In Fang et al. (2021) and Qiao et al. (2020), a separate pre-trained language model is used to enhance the language modeling capability.

Compared to previous techniques that concentrate solely on either visual or language representation learning, our proposed approach combines vision and language pretraining into a single, integrated model. This approach naturally integrates with the encoder-decoder recognition model, leading to improved representation quality for both vision and language.

**Vision-language pre-training for document understanding.** Some document understanding methods utilize vision-language pre-training to model both visual and linguistic information. For instance, the MVLM module in LayoutLM Xu et al. (2020) is proposed to align visual and textual representations, while VLPT Song et al. (2022) uses image-text contrastive learning to enhance visual representations.

Our proposed vision-language pre-training approach differs from LayoutLM and VLPT in several ways. Specifically, LayoutLM takes multi-modal information (visual tokens and textual tokens) as input to the network and learns to process both image and text information. In contrast, our approach only takes the image as input and utilizes language to assist in learning the decoder. Moreover, our approach differs from VLPT in that VLPT mainly uses masked language modeling and cross-model contrastive schemes to learn

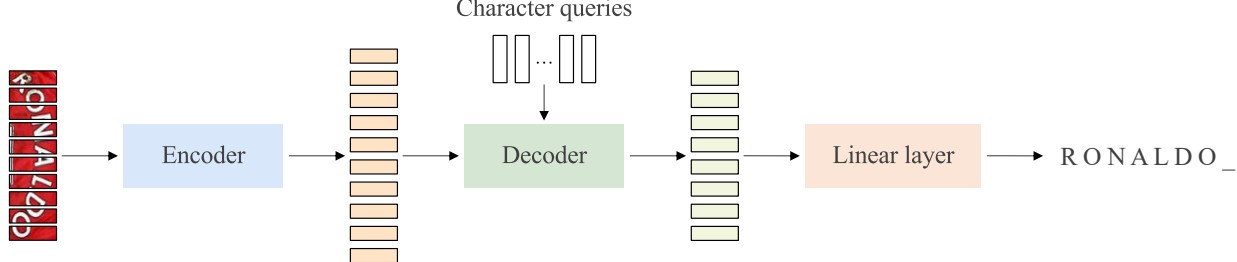

Figure 2: Encoder-decoder transformer for text recognition. The encoder extracts a sequence of patch representations, and the decoder maps the patch representations to a sequence of representations, followed by a linear layer to recognize the sequence of characters.

the image encoder, while our approach employs masked image modeling to learn the text-image encoder and language pre-training with masking for the decoder.

## 3 Approach

In this section, we will first present our encoder-decoder recognition model which is equipped with transformer. After that, the visual pre-training of encoder and the language pre-training of decoder will be described respectively.

### 3.1 Encoder-Decoder Transformer for Text Recognition

We adopt the encoder-decoder transformer architecture for text recognition. The encoder extracts a sequence of patch representations, and the decoder maps the patch representations to text representations. Figure 2 illustrates the encoder-decoder transformer architecture.

**Encoder.** The encoder receives an image, $\mathsf{I} \in \mathbb{R}^{3 \times H \times W}$, as the input. We partition the image vertically into a set of $M$ vertical patches,$[\mathsf{p}_1, \mathsf{p}_2, \ldots, \mathsf{p}_M]$. The size of each patch is $H \times W/M$. We process the flattened patches using linear projection to get patch embeddings. We add the 1D positional embeddings, which is enough as we partition the images vertically. We use the ViT Dosovitskiy et al. (2021), consisting of a sequence of multi-head self-attention and FFN units, as the encoder and learn the patch-level representations, $\mathbf{F} = [\mathbf{f}_1, \mathbf{f}_2, \ldots, \mathbf{f}_M]$, for the text image.

**Decoder.** We form the text recognition decoder by following the decoder style of the DETR Carion et al. (2020) that is originally designed for object detection. The decoder transforms the $N$ input embeddings, $\mathbf{C} = [\mathbf{c}_1, \mathbf{c}_2, \ldots, \mathbf{c}_N]$, called character queries, into output embeddings in parallel, which are then independently mapped into characters through a linear classifier.

**Loss.** The labels of the text recognition task are ordered strings. So unlike the DETR which uses the matching mechanism to assign ground truth for each output, we directly allocate the label of each character prediction in order. We denote the character predictions by $\mathbf{Y} = [\mathbf{y}_1 \ \mathbf{y}_2 \ \ldots \ \mathbf{y}_N]$. Assuming $N$ is larger than the number of characters in the text image. We consider the ground truth as $\mathbf{Y}^* = [\mathbf{y}_1^* \ \mathbf{y}_2^* \ \ldots \ \mathbf{y}_N^*]$ padded with an end-of-sentence symbol [EOS]. The loss function is formulated as follows,

$$\ell(\mathbf{Y}, \mathbf{Y}^*) = \frac{1}{L+1} \sum_{l=1}^{L+1} \mathrm{CE}(\mathbf{y}_l, \mathbf{y}_l^*), \tag{1}$$

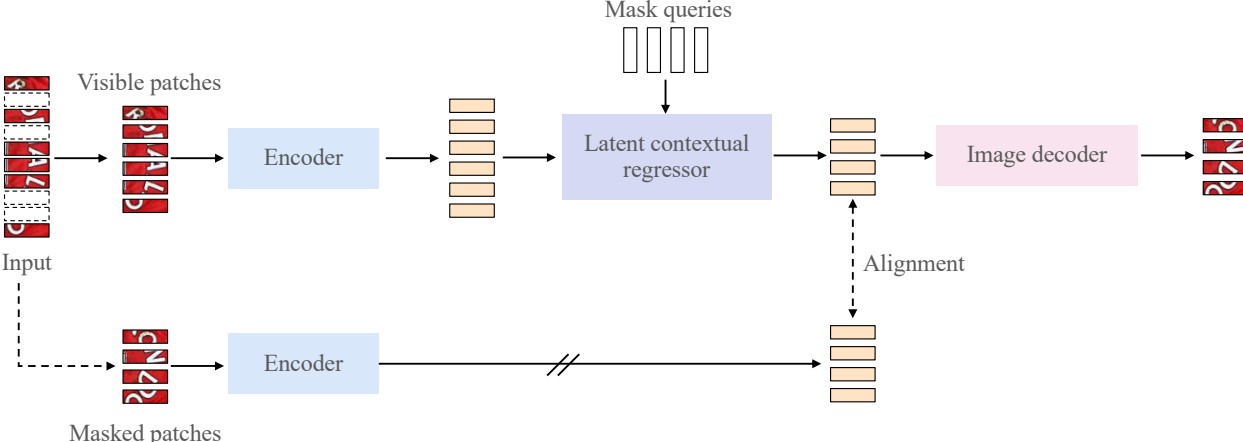

Figure 3: Visual pre-training on encoder. We adopt a masked image modeling approach to pre-train the encoder for text image representation learning. In this example, six image patches (top) are visible patches, and the other four (bottom) to be predicted are masked patches.

where $CE(\cdot, \cdot)$ is the cross-entropy loss. $L$ is the number of characters in the text image. To balance the number of [EOS] and other characters, we only employ the loss function on the characters as well as the first [EOS].

### 3.2 Masked Image Modeling on Encoder for Visual Pre-training

We follow the masked image modeling framework, which is recently studied for general image representation pre-training, and adopt the context autoencoder-style method Chen et al. (2022) to pre-train the encoder for text image representation learning. The context autoencoder separates the decoding role from the encoder and drives the encoder to focus on representation learning, which is a better choice for our visual representation learning.

The encoder pre-training process is given as follows. The input text image is randomly masked with a given masking ratio and the remaining parts, which are termed visible patches are sent to the encoder, generating the representations of visible patches. Then, the representations of visible patches are fed into a latent contextual regressor with mask queries, predicting the representations for masked patches $\mathbf{Z}_m$ which is expected to be close to the representations $\mathbf{Z}_m^*$ of masked patches directly computed from the encoder. Last, the representations of the predicted masked patches go into the image decoder, predicting the targets $\mathbf{T}_m$. Figure 3 illustrates the encoder pre-training architecture.

We use the patch RGB values, processed with layer normalization (Gaussian normalization), to form the targets. The loss function for encoder pre-training is a combination of representation alignment loss and prediction loss and is given as follows,

$$\ell_t(\mathbf{T}_m, \mathbf{T}_m^*) + \lambda \ \ell_z(\mathbf{Z}_m, \mathbf{Z}_m^*). \tag{2}$$

Here, both losses $\ell_t(\cdot, \cdot)$ and $\ell_z(\cdot, \cdot)$ are the MSE loss. $\lambda$ is the trade-off parameter and is set to 0.05 in our implementation.

### 3.3 Masked Image-Language Modeling on Decoder for Language Pre-training

Unifying the vision and language pre-training in a single model is not easy, since the granularity of image and text are different and the semantic of vision and language are not aligned. The core designs of our method to overcome this problem are that: 1) we transfer the text to image via text to image generation, thus the modality difference between image and text is eliminated. 2) We propose masked image-language modeling,

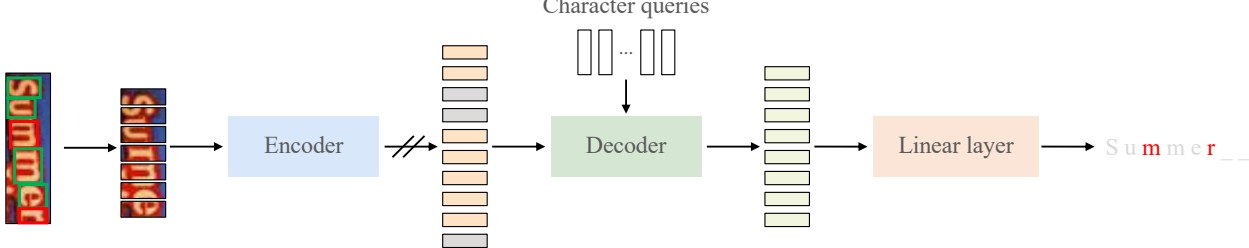

Figure 4: Language pre-training on decoder. The whole pipeline is similar with the one in Figure 2. The difference is that the input to the encoder are the visible patches. The visible patches are formed by masking the image patches that correspond to the target characters (the patch number may be greater than the character number). The input representations to the decoder are a combination of encoded representations and zero representations added to the positions (gray boxes) where the masked patches are. The prediction targets are the characters that are masked. The encoder is fixed when performing decoder pre-training.

which randomly masks some characters of the input image, and predicts the masked characters via the unmasked part. 3) We also design a sequential pre-training mechanism by freezing the pre-trained encoder to handle the domain gap between real image and synthesized image, so that the language presentation is enhanced while the pre-trained encoder which learns better visual representation is not affected.

In detail, we transform the text data to images via a public synthesis tool Text Render [1]. Given a text corpus, some fonts, and some background images, the synthesized text images as well as the location annotation of characters can be generated.

To further enhance the language modeling capability of the decoder, we adopt the idea of masked language modeling and introduce a masked image-language modeling scheme. As shown in Figure 4, we randomly mask some characters and accordingly the patches, and send the remaining visible patches to the encoder, obtain the representations of visible patches, $\mathbf{F}_v$. Then, we insert the zero representations $\mathbf{F}_m$ to the positions corresponding to the masked patches, and feed the combined representations $\mathbf{F} = [\mathbf{F}_v \ \mathbf{F}_m]$ with the corresponding positional embeddings added into the decoder, predicting the text: $\bar{\mathbf{Y}} = [\bar{\mathbf{y}}_1 \ \bar{\mathbf{y}}_2 \ \ldots \ \bar{\mathbf{y}}_N]$.

The loss function is similar to BERT Devlin et al. (2019) and is merely about masked characters. We formulate it as follows,

$$\ell(\bar{\mathbf{Y}}_l, \mathbf{Y}_l^*) = \frac{1}{L} \sum_{l=1}^{L} \mathrm{CE}(\mathbf{y}_{n_l}, \mathbf{y}_{n_l}^*), \tag{3}$$

where $L$ is the number of masked characters, and $\{n_1, n_2, \ldots, n_L\}$ are the positions of the masked characters.

Considering the style of synthesized text images might be different from the real text images, we keep the encoder (learned from encoder pre-training) not updated and only optimize the decoder, so that the pre-training stage of the decoder does not affect the representation quality of the pre-trained encoder.

## 4 Experiment

### 4.1 Datasets

**Chinese text line images.** The pre-training set consists of 100 million unlabeled text line images collected from practical scenarios for visual pre-training, and 100 million synthetic text line images for language pre-training. The real images are collected from documents and street view, and the text in them is almost in Chinese. We collect text corpus from Chinese corpus [2], and generate 100 million images with 64 commonly used fonts using Text Render. Specifically, for each synthetic sample, the text transcription as well as the character bounding boxes are given.

---

[1] https://github.com/oh-my-ocr/text_renderer
[2] https://github.com/crownpku/awesome-chinese-nlp

We first pre-train the encoder and decoder serially on the collected real images and the synthetic images, and then finetune our model on a large-scale Chinese text image benchmark BCTR Chen et al. (2021b). BCTR consists of four subsets (scene, web, document, and handwriting) and provides 1.4 million fully labeled images in total. The scene subset (Sen) is derived from some scene text datasets, including RCTW Shi et al. (2017b), ReCTS Zhang et al. (2019), LSVT Sun et al. (2019), ArT Chng et al. (2019), and CTW Yuan et al. (2019), resulting in 636,455 images. The web subset (Web) is constructed based on the MTWI He et al. (2018) dataset and contains 140589 text images. The document subset (Doc) is composed of 500000 synthetic text images generated by Text Render in document style. The handwriting subset (Hand) is collected from a handwriting dataset SCUT-HCCDoc Zhang et al. (2020), and 116643 text images are included.

**English text word images.** We follow Yang et al. (2022) and collect about 15.8 million unlabeled English word images from CC-OCR Yang et al. (2021) for visual pre-training. In addition, we also synthesize 100 million English word images for language pre-training. Similarly, we collect corpus from WikiText103 Merity et al. (2017) and generate synthetic images with Text Render and 10 commonly used English fonts.

Following Shi et al. (2019); Yu et al. (2020); Fang et al. (2021); Wang et al. (2021c); Zhang et al. (2022), two synthetic datasets MJSynth Jaderberg et al. (2014a) and SynthText Gupta et al. (2016) are used for the training of downstream recognition tasks. Besides, we also collect 2.78 million real labeled images from TextOCR Singh et al. (2021) and Open Images Dataset v5 [3] as Yang et al. (2022) to verify the effectiveness of pre-training when finetuned on real images. We evaluate our model on six public scene text datasets: ICDAR 2013 (IC13) Karatzas et al. (2013), Street View Text (SVT) Wang et al. (2011), IIIT5K-Words (IIIT5K) Mishra et al. (2012)), ICDAR 2015 (IC15) Karatzas et al. (2015), Street View Text-Perspective (SVTP) Phan et al. (2013), and CUTE80 (CUTE) Risnumawan et al. (2014)). The samples in the first three datasets are all regular text images and the remaining datasets may contain perspective or curved text images.

### 4.2 Implementation Details

**Encoder-decoder transformer.** The image patches are fed into a linear projection layer and then sent to the ViT. Two ViT structures are studied: ViT-S (12 transformer blocks with dimension 384), and ViT-B (12 transformer blocks with dimension 768). The decoder consists of four decoder layers, each of which includes a self-attention unit, a cross-attention unit, and an FFN unit. Each attention module is a 12-head attention with dimension 384.

We train the encoder-decoder transformer with AdamW optimizer Loshchilov & Hutter (2019), cosine learning rate decay Ilya Loshchilov and Frank Hutter (2017), a weight decay of 0.05, a drop path ratio of 0.1, and a batch size of 512. When the model is trained from scratch, the learning rate is set to $1e-3$. Otherwise, the model is optimized with an initial learning rate of $1e-4$. We set the training epochs as 120 and 20 for the Chinese text line recognition model and the English word recognition model with a warm-up of 5 epochs and 0.5 epochs respectively.

**Visual pre-training on encoder.** The latent contextual regressor consists of four regressor layers. Each layer includes a cross-attention unit and an FFN unit. The image decoder consists of four layers, and each layer includes a self-attention unit and an FFN unit. Each attention module is also a 12-head attention with dimension 384. Following He et al. (2022), we use the normalized pixel values of each masked patch as task.

We optimize the model with AdamW optimizer and set the learning rate with the linear learning rate scaling rule Goyal et al. (2017): $lr = base\_lr \times batchsize/256$. By default, the $base\_lr$ is set to $1.5e-4$ with cosine learning rate decay and a 0.5 epoch warm-up. We train the encoder for 10 epochs and 30 epochs for Chinese data and English data pre-training, with the batch size being 4096.

**Language pre-training on decoder.** We mask some characters with a ratio of 0.15 and accordingly mask the patches that contain the characters. This might lead to that a different number of patches are masked for

---

[3]https://storage.openvinotoolkit.org/repositories/openvino_training_extensions/datasets/open_images_v5_text

Table 1: Ablation about vision-language pre-training. "Scratch" means the model is trained from scratch. "V" and "L" mean visual pre-training and language pre-training respectively.

|          | Sce      | Web      | Doc      | Hand     | Avg      |
|----------|----------|----------|----------|----------|----------|
| Scratch  | 68.8     | 70.7     | 98.6     | 49.4     | 75.8     |
| V        | 72.3     | 73.7     | 99.2     | 62.5     | 79.8     |
| L        | 71.0     | 72.4     | 98.8     | 54.5     | 77.7     |
| V + L    | **73.9** | **74.8** | **99.3** | **63.7** | **80.8** |

different text images as one character may correspond to a different number of patches. We adopt masked attention to replace the original attention in the encoder with the parameters unchanged.

We pre-train the decoder for 5 epochs with a batch size of 512, an initial learning rate of $1e-4$, a 0.5 epochs warmup, and a cosine learning rate decay.

**Data preprocessing.** Since the Chinese text line images vary greatly in width, we resize the height of the input image to 32 with the aspect ratio kept and pad the width of the input images to 400. For the English word samples, we directly resize all input images to $32 \times 128$. We set the width of the split vertical patch to 4 for all datasets by default. During the training of downstream recognition, some data augmentations like rotation, distortion, and color jitter are also used.

**Evaluation** We evaluate BCTR by first processing the predictions and ground truth with the following rules as Chen et al. (2021b): (i) convert the full-width characters to half-width characters; (ii) convert all traditional Chinese characters to simplified characters; (iii) convert all English characters to lowercase; (iv) remove all spaces. After that, we compute the accuracy in sentence level over each subset and the whole dataset (Avg).

To evaluate the six scene English text datasets, we follow Shi et al. (2019); Yu et al. (2020); Fang et al. (2021); Wang et al. (2021c); Zhang et al. (2022) and evaluate the recognition performance of our model with case-insensitive word accuracy. We also report the average accuracy (Avg) over all samples.

### 4.3 Ablation Studies

In this section, we conduct ablation studies on the BCTR dataset to verify the effectiveness of our proposed pre-training method. All experiments are conducted on 8 A100 GPUs with the ViT-B as the encoder.

**The effectiveness of vision-language pre-training.** To verify the effectiveness of vision-language pre-training, four models are trained. (i) The model is randomly initialized. (ii) The encoder is initialized with visual pre-training, and the rest parts are randomly initialized. (iii) The model is initialized with language pre-training which pre-trained on synthetic data. (iv) The model is initialized with vision-language pre-training.

The results are presented in Table 1, which demonstrate the effectiveness of vision-language pre-training. Specifically, the superiority of the "V" model over the "Scratch" model indicates that visual pre-training of the encoder can lead to an improvement of up to 4%. In addition, language pre-training also yields better performance, achieving a 1.9% improvement over the "Scratch" model. Moreover, the visual pre-training and language pre-training are complementary, as evidenced by the 1% improvement achieved by the "V + L" model over the "V" model. These results provide strong evidence that language pre-training is a valuable approach.

**Evaluation of representation quality.** To evaluate the quality of representations learned by our pre-trained model, we conducted linear probing experiments and the results are shown in Table 2. We fixed the pre-trained encoder and decoder, which was pre-trained using vision-language pre-training, and only updated the parameters of the remaining linear layer. Surprisingly, the model achieved an average accuracy

Table 2: Evaluation of representation quality. We consider two cases : (i) to evaluate the representation quality of the encoder which with vision pre-training, we fix the pre-trained encoder and finetune the remain part of the recognition model. we termed this setting as "V Linear probing". (ii) We conduct "V + L linear probing" that fixes the pre-trained encoder and decoder to evaluate the representation quality of vision-language pre-training. Only the remaining linear layer is updated in this setting.

|  | Sce | Web | Doc | Hand | Avg |
|---|---|---|---|---|---|
| V Linear probing | 62.8 | 67.5 | 98.1 | 54.3 | 73.6 |
| V + L Linear probing | 39.1 | 54.7 | 66.3 | 28.1 | 47.8 |

Table 3: Studying if the pre-trained encoder is simultaneously retrained during decoder pre-training. (i) The model is trained from scratch. (ii) the pre-trained encoder is retrained during the decoder pre-training. (iii) The pre-trained encoder is fixed when conducting decoder pre-training.

|  | Sce | Web | Doc | Hand | Avg |
|---|---|---|---|---|---|
| Scratch | 68.8 | 70.7 | 98.6 | 49.4 | 75.8 |
| Retrain encoder | 69.0 | 71.4 | 99.0 | 53.3 | 76.7 |
| Fix encoder | **73.9** | **74.8** | **99.3** | **63.7** | **80.8** |

of 47.8%, indicating that with vision-language pre-training, the encoder and decoder are already capable of learning meaningful representations.

We also examined the quality of representations learned through visual pre-training, which fixes the pre-trained encoder and fine-tunes the decoder and linear layer of the recognition model. The model achieved an average accuracy of 73.6%, demonstrating that visual pre-training also enhances the representation quality of the encoder.

**The effectiveness of our serially pre-training mechanism.** To address the large domain gap between real and synthesized images, we fixed the pre-trained encoder when conducting language pre-training on the decoder. A comparison of our serially pre-trained mechanism with the encoder retrained from pre-trained weights is shown in Table 3, which demonstrates the effectiveness of our approach. Specifically, when retraining the encoder from pre-trained weights during decoder pre-training, an average accuracy of 76.7% was achieved, which outperformed the scratch model (75.8%). However, this approach yielded a 4.1% drop in accuracy compared to our serially pre-trained model (80.8%), suggesting that the pre-trained encoder would be impacted by the synthesized text images if it were retrained during decoder pre-training. This emphasizes the large domain gap between synthetic and real data.

**Masking ratio of visual pre-training** We conducted an exploration into the effect of different masking ratios, namely 0.30, 0.45, and 0.60, for visual pre-training. The results have been presented in Table 4, and we found that the optimal masking ratio for downstream recognition task was 0.45. It is worth noting that our optimal masking ratio is lower than that reported in MAE He et al. (2022) (0.75). This discrepancy could be attributed to the higher information density of the text images used in our experiments.

Table 4: Ablation about the masking ratio of visual pre-training. In our experiments, the optimal masking ratio for downstream recognition task was 0.45.

| Masking ratio | Sce | Web | Doc | Hand | Avg |
|---|---|---|---|---|---|
| 0.30 | 71.5 | 73.1 | 99.1 | 61.8 | 79.3 |
| 0.45 | **72.3** | **73.7** | **99.2** | **62.5** | **79.8** |
| 0.60 | 72.0 | 73.6 | 99.1 | 60.7 | 79.4 |
| 0.75 | 72.2 | 73.4 | 99.1 | 59.1 | 79.2 |

Table 5: Ablation about the masking strategy in language pre-training. It can be seen that masking helps the performance, particularly for challenging scenarios such as scenes and handwritten text recognition, where accurate recognition is more difficult to achieve. "M" means trained with the masking strategy and "V" means trained with the visual pre-training.

| M | V | Sce | Web | Doc | Hand | Avg |
|---|---|---|---|---|---|---|
| × | × | 69.3 | 71.4 | 98.6 | 49.3 | 76.1 |
| √ | × | **71.0** | **72.4** | **98.8** | **54.5** | **77.7** |
| × | √ | 73.1 | 74.6 | 99.3 | 63.6 | 80.5 |
| √ | √ | **73.9** | **74.8** | 99.3 | **63.7** | **80.8** |

Table 6: Comparison with supervised pre-training. (i) The model is trained without pre-training model. (ii) The pre-trained model is trained on synthetic data with supervised pre-training. (iii) The model is pre-trained with our vision-language pre-training.

| | Sce | Web | Doc | Hand | Avg |
|---|---|---|---|---|---|
| Scratch | 68.8 | 70.7 | 98.6 | 49.4 | 75.8 |
| Supervised pre-training | 69.3 | 71.4 | 98.6 | 49.3 | 76.1 |
| Ours | **73.9** | **74.8** | **99.3** | **63.7** | **80.8** |

**Masking strategy of language pre-training** We have examined the impact of masking strategy on language pre-training in Table 5. The results show that the language models trained with the masking strategy outperformed those without on both the scenarios of only conducting language pre-training and integrating visual pre-training. The results highlight the efficacy of masking strategy in capturing the underlying linguistic rules during pre-training, particularly for challenging scenarios such as scenes and handwritten text recognition.

**Comparison with supervised pre-training.** To demonstrate the superiority of our vision-language pre-training, we conducted a comparison with supervised pre-training, the results of which are presented in Table 6. Specifically, we pre-trained the recognition model on 100 million synthetic images and fine-tuned it on the BCTR dataset. Our findings show that, compared to the model trained from scratch, the model with supervised pre-training only achieved marginal improvement, indicating that supervised pre-training with synthetic images is unnecessary when there is an abundance of real labeled data. However, our proposed vision-language pre-training resulted in a 5% improvement, highlighting its effectiveness.

**Vertical patch size.** In our study, we evaluated the performance of two different patch sizes without pre-training, the results of which are presented in Table 7. We found that the larger patch size led to worse performance, dropping the accuracy by 3.6%. We hypothesize that this may be due to the higher information density of the embedded token in the larger patch size, making it more difficult to learn.

Table 7: Ablation about the vertical patch size.

| Patch size | Sce | Web | Doc | Hand | Avg |
|---|---|---|---|---|---|
| $32 \times 4$ | **68.8** | **70.7** | **98.6** | **49.4** | **75.8** |
| $32 \times 8$ | 64.0 | 67.3 | 97.5 | 43.3 | 72.2 |

**Generalizability.** To demonstrate the effectiveness and generalizability of our proposed vision-language pre-training, we conducted experiments on a widely used text recognition decoder, the Connectionist Temporal Classification (CTC) decoder. We utilized multiple transformer encoder layers as the sequence decoder and trained the model using CTC loss. During language pre-training, we randomly masked some characters and predicted the entire text sequence using CTC loss.

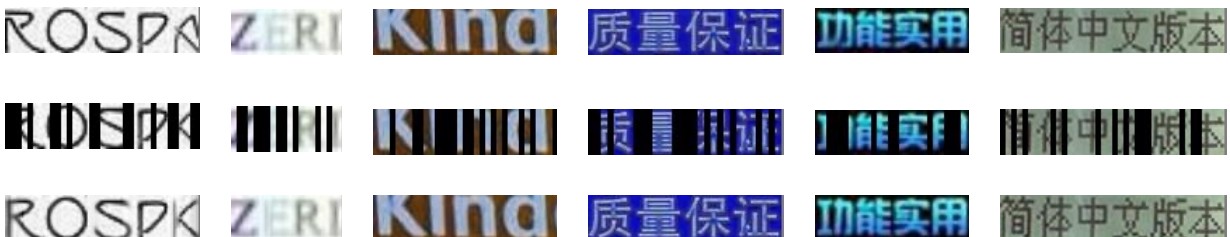

Figure 5: Example results of masked image modeling. The first row are the input images, the remaining two rows are masked images and reconstructed images.

| | oscr | oscar | | 扩建府目 | 扩建项目 |
| | vvilla | villa | | 毫元关系 | 毫无关系 |
| | piizza | pizza | | 木发铜锅 | 木炭铜锅 |
| | disnep | disney | | 质量是天键 | 质量是关键 |
| | strawbering | strawberry | | 吉乐工作室 | 声乐工作室 |

Figure 6: Visualization of text recognition results. The images in the first column and the fourth column are the input images. The results in the second column and the fifth column are predicted by the models trained from scratch. The results in column 3 and column 6 are output by the models with vision-language pre-training.

To validate the effectiveness of our approach, we trained two models: one trained from scratch and the other initialized with vision-language pre-training. Our results in Table 8 demonstrate that the model initialized with vision-language pre-training achieved higher accuracy, with a 3.4% performance gain over the scratch model. Specifically, we achieved accuracies of 76.7% and 80.1%, respectively. These findings support the robustness and generalizability of our proposed model in the encoder-decoder recognition framework.

Table 8: Ablation about the different decoders.

| Decoders | Scratch | Vision-language Pre-training |
| --- | --- | --- |
| CTC-style | 76.7 | 80.1 |
| DETR-style | 75.8 | 80.8 |

**Qualitative analysis.** We visualize some examples result of masked image modeling in Figure 5. The input images are randomly masked with a masking ratio of 0.45. As a result, a part of a character or a whole character may be not visible for the encoder. The semantically plausible reconstructed results in the third row show that meaningful visual representation are learned by the encoder.

We also show some results of the models without and with vision-language pre-training in Figure 6. As shown, when trained from scratch, the model is not robust to artistic fonts, occluded text, and visually similar characters. Benefiting from the visual and linguistic prior knowledge of real images and text corpus, the model with vision-language pre-training can handle the above-mentioned scenarios well.

## 4.4 Comparison with State-of-the-art Methods

**Comparison with state-of-the-art pre-training method.** We compare our method with previous state-of-the-art visual pre-training method DiG Yang et al. (2022) fairly with the same data setting. Specifically,

Table 9: Comparison with state-of-the-art pre-training method DiG. "Scratch" denotes the model is trained from scratch. "V" means finetuned with visual pre-training, "V + L" means finetuned with vision-language pre-training. The percentages 1%, 10%, and 100% refer to the proportions of real data used during the fine-tuning stage.

|                      | 1%   | 10%  | 100% | #Params |
|----------------------|------|------|------|---------|
| CRNN Scratch         | 63.5 | 79.6 | 89.4 | 25M     |
| DiG (ViT-S) Scratch  | 9.2  | 73.4 | 91.4 | 36M     |
| DiG (ViT-S) V        | 84.6 | 92.0 | 94.6 | 36M     |
| Ours (ViT-S) Scratch | 72.3 | 90.6 | 95.2 | 31M     |
| Ours (ViT-S) V       | 87.7 | 92.6 | 95.3 | 31M     |
| Ours (ViT-S) V + L   | **90.9** | **93.8** | **95.6** | 31M |

we first pre-train our model on CC-OCR, MJSynth, and SynthText, following Yang et al. (2022) and finetune the pre-trained model on varying percentages of real labeled samples (1%, 10%, and 100%) sourced from TextOCR and Open Images Dataset v5. Our model's performance was thoroughly evaluated and compared against DiG, as presented in Table 9.

We discovered that pre-training DiG led to a significant improvement in performance compared to the DiG baseline results obtained without pre-training. However, it is unclear whether this improvement is solely due to DiG pre-training or differences in the training settings. To reduce uncertainty, we trained a commonly used recognition model, CRNN Shi et al. (2017a)[4], using the same data settings as our model, and achieved results of 63.5%, 79.6%, and 89.4%. Notably, CRNN was published in 2015 and its performance is generally not competitive with most recent recognition methods. The results of CRNN in Table 9 are significantly higher than the DiG baseline, suggesting that the observed performance gains may be primarily due to differences in the training settings used for the DiG baseline rather than the DiG pre-training.

Due to the aforementioned reasons, we directly compared the performance of pre-training. Remarkably, our proposed approach significantly outperformed DiG in all the cases. Most notably, our method achieved an exceptional accuracy of 90.9% with only 27.8K (1%) real labeled images during finetuning, providing substantial evidence of the significant reduction in the demand for labeled data offered by our approach. Furthermore, the performance gains observed in our model, equipped with pre-trained encoder and decoder, further underscore the effectiveness of our pre-training mechanism.

**Chinese Text Line Recognition.** We evaluate the ability of our model to recognize Chinese text lines on the BCTR dataset. We set the number of character queries $N$ to 40 since most of the samples in BCTR have less than 40 characters. We show the results of our method and some representative methods on the BCTR dataset in Table 10. When training from scratch, our method with ViT-S as encoder outperforms all the previous methods while with the smallest model size. Specifically, our method is better than the previous best method TransOCR Chen et al. (2021a) by 2.8% (72.8% vs. 75.6%). When training with the pre-trained encoder and decoder, our models surpass the previous best results by large margins. In detail, our method shows steady improvement with the increase of the model size and improves over the state-of-the-art by 5.3% and 8.0% respectively.

**English scene text recognition.** Following Shi et al. (2017a; 2019), we set the number of character queries N to 25 which exceeds the lengths of most English words. Since scene text appeared in natural scenes always with distortions or irregular layouts, we employ a spatial transformer network Jaderberg et al. (2015) which is adopted in Shi et al. (2019) to rectify the input image and train it with our recognizer jointly.

Text recognition in the early days faced the challenge of limited annotated data, which led to the common practice of training on synthetic data and testing on real data. However, as data has accumulated over the past decade and unsupervised techniques have advanced, pre-training on synthetic data and testing

---

[4]https://github.com/PaddlePaddle/PaddleOCR

Table 10: Text recognition results on the BCTR dataset. VP and LP mean Visual Pre-training and Language Pre-training respectively.

| Methods | VP | LP | Sce | Web | Doc | Hand | Avg | #Params |
|---|---|---|---|---|---|---|---|---|
| CRNN Shi et al. (2017a) | × | × | 53.4 | 54.5 | 97.5 | 46.4 | 67.0 | - |
| ASTER Shi et al. (2019) | × | × | 54.5 | 52.3 | 93.1 | 38.9 | 64.7 | - |
| SAR Li et al. (2019) | × | × | 62.5 | 54.3 | 93.8 | 31.4 | 67.3 | - |
| SRN Yu et al. (2020) | × | × | 60.1 | 52.3 | 96.7 | 18.0 | 65.0 | - |
| TransOCR Chen et al. (2021a) | × | × | 63.3 | 62.3 | 96.9 | 53.4 | 72.8 | 84M |
| **Ours (ViT-S)** | × | × | 68.6 | 70.3 | 98.5 | 49.1 | 75.6 | 36M |
| **Ours (ViT-B)** | × | × | 68.8 | 70.7 | 98.6 | 49.4 | 75.8 | 100M |
| **Ours (ViT-S)** | √ | √ | 71.4 | 72.5 | 98.8 | 55.6 | 78.1 | 36M |
| **Ours (ViT-B)** | √ | √ | **73.9** | **74.8** | **99.3** | **63.7** | **80.8** | 100M |

Table 11: Text recognition results on six English scene text datasets. VP, LP and AD mean Visual Pre-training, Language Pre-training and Annotated Data respectively.

| Methods | VP | LP | AD | IC13 | SVT | IIIT5K | IC15 | SVTP | CUTE | Avg | #Params |
|---|---|---|---|---|---|---|---|---|---|---|---|
| ASTER Shi et al. (2019) | × | × | synth | 91.8 | 89.5 | 93.4 | 76.1 | 78.5 | 79.5 | 86.7 | - |
| TextScanner Wan et al. (2020) | × | × | synth | 92.9 | 90.1 | 93.9 | 79.4 | 84.3 | 83.3 | 84.4 | - |
| PIMNet Qiao et al. (2021) | × | × | synth | 95.2 | 91.2 | 95.2 | 83.5 | 84.3 | 84.4 | 90.5 | - |
| SRN Yu et al. (2020) | × | × | synth | 95.5 | 91.5 | 94.8 | 82.7 | 85.1 | 87.8 | 90.4 | 55M |
| VisionLan Wang et al. (2021c) | × | × | synth | 95.7 | 91.7 | 95.8 | 83.7 | 86.0 | 88.5 | 91.2 | 33M |
| ABINet Fang et al. (2021) | × | × | synth | 94.9 | 90.4 | 94.6 | 81.7 | 84.2 | 86.5 | 89.8 | 24M |
| I2C2W Xue et al. (2021) | × | × | synth | 95.0 | 91.7 | 94.3 | 82.8 | 83.1 | 93.1 | 90.2 | - |
| **Ours (ViT-S)** | × | × | synth | 97.7 | 93.7 | 95.4 | 86.6 | 89.0 | 87.5 | 92.5 | 31M |
| **Ours (ViT-B)** | × | × | synth | 96.8 | 94.7 | 95.3 | 87.1 | 89.3 | 90.6 | 92.7 | 97M |
| SEED Qiao et al. (2020) | × | √ | synth | 92.8 | 89.6 | 93.8 | 80.0 | 81.4 | 83.6 | 88.3 | - |
| ABINet Fang et al. (2021) | × | √ | synth | 97.4 | 93.5 | 96.2 | 86.0 | 89.3 | 89.2 | 92.7 | 37M |
| ConCLR Zhang et al. (2022) | × | √ | synth | 97.7 | 94.3 | 96.5 | 85.4 | 89.3 | 91.3 | 92.8 | 37M |
| PerSec Liu et al. (2022) | √ | × | synth | 97.2 | 94.6 | 96.3 | 84.4 | 89.5 | 90.2 | 92.4 | - |
| DiG (ViT-S) Yang et al. (2022) | √ | × | synth | 97.1 | 93.4 | 96.7 | 87.1 | 90.1 | 88.5 | 93.1 | 36M |
| DiG (ViT-B) Yang et al. (2022) | √ | × | synth | 96.9 | 94.6 | 96.7 | 87.1 | 91.0 | 91.3 | 93.4 | 52M |
| TrOCR Li et al. (2021b) | √ | √ | synth | **98.3** | 93.2 | 91.0 | 84.0 | 91.0 | 89.6 | 90.3 | 558M |
| **Ours (ViT-S)** | √ | √ | synth | 97.7 | 94.0 | 95.8 | 87.5 | 90.2 | 89.2 | 93.0 | 31M |
| **Ours (ViT-B)** | √ | √ | synth | 98.1 | 94.9 | 95.8 | 87.5 | 89.8 | 90.3 | 93.1 | 97M |
| DiG (ViT-S) Yang et al. (2022) | √ | × | real | 97.3 | 96.1 | 97.7 | 88.6 | 91.6 | 96.2 | 94.6 | 36M |
| DiG (ViT-B) Yang et al. (2022) | √ | × | real | 97.6 | 96.5 | 97.6 | 88.9 | 92.9 | **96.5** | 94.9 | 52M |
| MAERec-S Jiang et al. (2023) | √ | × | real | - | - | - | - | - | - | 94.6 | 21M |
| **Ours (ViT-S)** | √ | √ | real | 97.8 | **96.9** | **98.0** | **90.2** | 94.9 | 96.2 | **95.6** | 31M |
| **Ours (ViT-B)** | √ | √ | real | 98.2 | **96.9** | **98.0** | 90.1 | 94.6 | 95.8 | **95.6** | 97M |

on real data has become increasingly impractical. Firstly, there is now an abundance of millions of real data samples available, providing sufficient resources for training recognition models. Secondly, domain differences between synthetic and real data can cause differences in the effectiveness of training. Therefore, the performance demonstrated in synthetic data training may not be representative of real data training. Thirdly, current unsupervised methods usually pre-train on real data to learn good feature representations, and fine-tuning on synthetic data may impair the effectiveness of pre-training. Consequently, we advocate pre-training on unlabeled real data and fine-tuning on labeled real data for evaluating model performance. As most methods are trained only on synthetic data, we also provide a comparison with this approach for reference. We report the results of our method and existing methods in Table 11 and compare them in detail.

When trained on "synth" (MJSynth + SynthText) without pre-training, our method significantly surpasses the previous methods and achieves the best performance. When trained on synthetic data with pre-training, our method achieves comparable results with previous best model DiG Yang et al. (2022). Besides, When

Table 12: Comparison of run time. FPS is an abbreviation for Frames per Second, which denotes the number of images processed per second.

| Methods | Avg | #Params | Flops | FPS |
|---|---|---|---|---|
| ABINet Fang et al. (2021) | 92.7 | 37M | 2.97G | 545 |
| DiG Yang et al. (2022) | 94.6 | 36M | 15.74G | 188 |
| Ours(ViT-S) | **95.6** | **31M** | **1.17G** | **1160** |

finetuned on real datasets (TextOCR + Open Images Dataset v5) as Yang et al. (2022); Jiang et al. (2023), our method achieves the best performance (95.6% vs. 94.9% vs. 94.6%). The extensive comparisons demonstrate the excellence of our proposed method.

Note that, some methods may also achieve good performance but not relevant to pre-training are not listed. These methods, perform semi-supervised learning on real data Fang et al. (2021); Zheng et al. (2022), refine the results with iterative mechanism Fang et al. (2021); Bautista & Atienza (2022), or predict text in multi-granularity Wang et al. (2022) are complementary to our approach and may benefit our approach.

We also observe that the improvement brought by pre-training is more pronounced for Chinese benchmarks than for English benchmarks. This could be attributed to several reasons. Firstly, the current English recognition datasets contain fewer characters and shorter text length, making it relatively easier than Chinese recognition. Secondly, the accuracy of current models on English benchmarks is already high and near saturation, with only marginal gains from pre-training as shown by the small improvement margin (0.4%) between DiG (ACM MM2022) and ABINet (CVPR 2021). Lastly, pre-training is more beneficial for challenging scenarios where the original model performs poorly. In many real-world scenarios that are more challenging and lack sufficient labeled data, the performance gain from pre-training can be substantial, such as the handwriting set of BCTR.

**Runtime Comparison.** We compare the runtime of our method with two competitive other recognition methods that with similar parameters as ours. The runtime is evaluated on an A100 GPU with a batch size of 64, the results are shown in Table 12. Compared to ABINet and DiG, our method has the smallest number of parameters, the highest accuracy, and the fastest running speed. We measured the computational complexity of the three networks to analyze the difference in inference speed. ABINet has 2.97G Flops, DiG has 15.74G Flops, and MaskOCR has 1.17G Flops, showing a positive correlation with speed. We further investigated why DiG's computational complexity far exceeds that of ABINet and MaskOCR under similar parameters. We found that DiG uses a relatively small patch size (4 * 4), while our approach employs a patch size of 32 * 4. This results in a longer feature sequence input to ViT in DiG, leading to an exponential increase in computational complexity.

## 5 Conclusion

The core of the proposed approach for text recognition lies in that we pre-train the recognition model, including both the encoder and the decoder to learn vision and language representation. The visual pre-training is able to benefit from large-scale real text images that are easily available without the need for text annotation. The language pre-training is able to benefit from the synthetic text images that are also easily available with the character-level annotation easily obtained. Experiments verify the effectiveness of our proposed vision-language pre-training.

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
