# OpenReview forum: "MaskOCR: Scene Text Recognition with Masked Vision-Language Pre-training"
_TMLR — Accepted by TMLR_

### Review · Reviewer_jHaA · 2023-12-07

**Summary Of Contributions:**

This paper discusses a pretraining strategy for encoder-decoder-based OCR systems and proposes a training procedure consisting of three steps: Pretraining of encoder with real images, pretraining of decoder with synthetic images while fixing the encoder part, and finetuning of the whole system. The pretraining criteria are basically the prediction of masked regions of pixels and texts for encoder and decoder, respectively. The key design principle is that the pretraining of encoder is done by real images and the pretraining of decoder is done by synthetic images while freezing the encoder. It allows to pretrain all the components effectively. The extensive experiments show the effectiveness of the approach and it reports strong numbers on OCR benchmarks in English and Chinese.

**Audience:**

Yes

**Broader Impact Concerns:**

No concerns.

**Claims And Evidence:**

Yes

**Requested Changes:**

It seems the conclusion from the results in Table 1, 2, 5 is that we should never train the encoder part with synthetic data only. Does that sound reasonable? What if we combine real images with synthetic data when finetuning, etc. It would be great to have discussions on this.

There are many tables with the same result at the first row (i.e. Scratch). I'm wondering if it is a better idea to create a consolidated table listing all results. There are several numbers that are just mentioned in text. Those could be put in the table, too. If it makes the readability worse, you can ignore this.

There seem to be many unnatural expressions in the text although I was able to understand the meaning. I would recommend a proof-reading by a third-party person.

**Strengths And Weaknesses:**

The key contributions of the paper are 1) the use of real images for pretraining of encoder, 2) the use of synthetic images for pretraining of decoder while freezing encoder, 3) both encoder and decoder are pretrained. None of which is probably very novel by itself, but the combination and providing the strategy supported by the thorough experiments have a good value. Especially that showing a practical strategy to pretrain decoder with synthetic images seems important. It gives an insight how to use synthetic images for training OCR systems. The experiments are extensive and it is great that the methods are evaluated on two languages.

The presentation could be improved in terms of writing and how to summarize results, but I do not find major weakness in the paper. A possible weakness is that the comparisons with prior studies may not be very fair or it may not contrast the differences in terms of algorithm. It is a shared common problem in the field that you just put the numbers from other papers. As discussed in the paper, we often observe significantly better results by just retraining a simple weak model like Resnet. It is great to have results of GiG from this perspective.

---

> ### Author Response · Authors · 2024-02-27
> **Response to reviewer jHaA**
>
> Thank you for the review and the valuable comments/suggestions! We have attempted to address them in the responses below.
>
> **"combine real images with synthetic data when finetuning"**
>
> This is a valuable insight. In fact, our intention to fix the encoder is to prevent the pre-trained encoder, optimized on real data, from being influenced by synthetic data. Therefore, from this perspective, combining real and synthetic data during the pre-training phase for the decoder seems to be a convergent strategy. We will explore this approach, as well as the pre-training of both encoder and decoder in a single stage, in our future work.
>
> **"There are several numbers that are just mentioned in text. Those could be put in the table, too."**
>
> Thank you for your suggestion. We will reorganize the experimental results described in the text into tables to enhance the readability of the paper.
>
> **"There seem to be many unnatural expressions in the text "**
>
> "Thank you for your suggestion. We will carefully revise our expressions to make them read more naturally."

---

### Review · Reviewer_gMeq · 2024-01-27

**Summary Of Contributions:**

This paper presents a novel approach to text recognition that combines vision and language pre-training in a classical encoder-decoder framework. The authors propose masked image modelling for the encoder and masked image-language modelling for the decoder. The approach involves pre-training the encoder on unlabeled real text images and the decoder on synthesized text images, thereby enhancing both visual and linguistic representation learning. The paper presents extensive experiments on Chinese and English text images, demonstrating superior performance over state-of-the-art methods.

**Audience:**

No

**Claims And Evidence:**

No

**Requested Changes:**

- In section 3.1, it is written "We partition the image vertically into a set of M vertical patches,[p1, p2, . . . , pM]." This supposes that the text orientation on the image is given.
- The loss (Eq. 1) is purely sequential, which implies that a single insertion makes the full sequence considered incorrect. It is surprising that the model can be trained with this loss whereas loss based on alignment such as CTC have been generally used in the document recognition community.
- Section 4.2 Evaluation: the metrics used for BCTR is the sentence level accuracy, which seem very strict (one character error discard the full sentence). Could you confirm? The metrics is different for the English datasets.
- Table 3: the model is not very sensitive to the masking ration, which is good. What are the results with a masking ratio at 0.75 which is reported as the best for MAE?
- Table 7: what are the 1% 10% 100% for?
- Table 10: what is the time unit?

**Strengths And Weaknesses:**

First, this paper, entitled "Text Recognition with Masked Vision-Language Pre-training", focuses primarily on the area of scene text recognition, a detail that could be more explicitly reflected in the title for clarity. While the paper presents a method that integrates vision and language pre-training within an encoder-decoder architecture, it notably omits any discussion of various models developed for document recognition, which are closely related to the field of scene text recognition. Furthermore, the related work section would benefit greatly from a more explicit exploration of the links between these two fields.

Second, The approach presented in the paper, which integrates vision and language pre-training within an encoder-decoder architecture for text recognition, must be reconsidered in the context of the paper introducing the TrOCR model, particularly its 2023 version as presented at the Thirty-Seventh AAAI Conference on Artificial Intelligence (AAAI-23). It's noteworthy that while the paper cites the 2021 version of TrOCR, it overlooks the significantly updated and improved 2023 iteration. This oversight raises concerns regarding the originality and contemporary relevance of the proposed approach.
Indeed, the core of the paper is centered on the proposed pretraining methodology, which 1) combines text-to-image conversion, 2) masked image-language modeling, and 3) a strategic use of a frozen encoder to effectively bridge the gap between visual and textual data interpretation.
The first and second proposition are already used in the 2021 TrOCR paper. The third proposition seems to be new compared to trOCR and to improve the results as show on table 2. However, the main contribution of this article comes down to this aspect, which is very limited.

Third, in this text recognition model, there is a necessary dependency on a separate text recognition model to provide the positions of the text, a dependency that should be explicitly stated. It's important to anticipate a potential drop in performance when these two models are combined, as they are not optimized together. This raises a critical question: given the interdependencies and potential performance trade-offs, is it still worth optimizing the text recognition module in isolation? This broader perspective should be considered when assessing the overall effectiveness and progress of approaches in this area.

It's worth noting that there is an emerging trend within the document recognition community towards the development of fully integrated models that combine both text detection and recognition(End-to-End Document Recognition and Understanding with Dessurt in ECCV2022, DAN : a Segmentation-free Document Attention Network for Handwritten Document Recognition in PAMI 2023). The field's shift towards integrated models highlights the importance of addressing the issues of interdependence and potential performance trade-offs when text detection and recognition processes are not optimized together.

---

> ### Author Response · Authors · 2024-02-27
> **Response to reviewer gMeq**
>
> Thank you for the review and the valuable comments/suggestions! We have attempted to address them in the responses below.
>
>
> **Concern about the title**
>
> Thank you for your suggestion. We will modify the title of the paper from 'Text Recognition with Masked Vision-Language Pre-training' to 'Scene Text Recognition with Masked Vision-Language Pre-training' for clarity.
>
> **Comparsion with TrOCR**
>
> Our approach shares the same objective as TrOCR, aiming to enhance text recognition through pre-training.  However, there are distinct differences in our methodologies. TrOCR employs an auto-regressive encoder-decoder framework, enhancing model performance through pre-training on synthetic data (after carefully reviewing the 2023 version of TrOCR, we found no evidence suggesting that TrOCR also utilizes the masked image-language modeling method). In contrast, we utilize the masked image modeling approach to pre-train the feature encoder using a substantial set of unlabeled real text images, allowing us to acquire robust visual representations. Additionally, we enhance the language modeling capability of the sequence decoder through a proposed masked image-language modeling scheme applied to synthetic data.
>
> **Text recognition v.s. end-to-end text recognition.**
>
> We agree with the reviewer's insight that end-to-end optimization might offer greater potential for optimizing both detection and recognition models in the future. However, we argue that, at present, researching independent models remains valuable for several reasons:
>
> Flexibility: End-to-end optimization requires datasets annotated for both detection and recognition, which are relatively scarce. In contrast, datasets with only detection or recognition labels are more abundant and easier to collect. Additionally, optimizing detection and recognition models independently provides greater flexibility due to differing data requirements and iteration steps during training.
>
> Efficiency: In practical applications, end-to-end models may encounter efficiency constraints. For instance, to effectively recognize small text, end-to-end models might require resizing images to larger resolutions. Independent models, on the other hand, can use lower resolutions for text detection and subsequently obtain accurate recognition on high-resolution Regions of Interest (ROIs). This results in efficiency advantages during deployment.
>
> Moreover, recent text recognition competition winners, such as ReST and DSText, still utilize separate models for detection and recognition. This observation reinforces the ongoing value of researching independent models at this stage. While we acknowledge that end-to-end solutions may overcome these challenges with technological advancements, we will continue exploring this direction in our future work.
>
> **"This supposes that the text orientation on the image is given..."**
>
> Yes, similar to the majority of previous recognition methods, we assume that the input text lines are oriented horizontally. However, this does not imply that our method is incapable of recognizing vertically oriented text. Simply rotating the vertical text by 90 degrees, allowing it to be recognized effectively just like horizontal text.
>
> **"The loss (Eq. 1) is purely sequential..."**
>
> The loss (Eq. 1) is widely applied in tasks involving sequence generation, including but not limited to speech recognition, machine translation, and OCR. Moreover, the effectiveness of this loss has been validated by numerous text recognition models(e.g. ASTER, SRN, ABINet, DIG et al.). Additionally, the framework we propose is versatile, applicable not only to the loss (Eq. 1) but also to the CTC loss. We have provided a comparative validation of these two losses in the "Generalizability" section on page ten of the submitted paper.
>
> **"the metrics used for BCTR"**
>
> Yes, we confirm. Sentence-level accuracy is among the metrics used in BCTR, denoted as ACC in the BCTR paper. In essence, if we consider English words as a sentence, these two terms are equivalent, with the only distinction being the difference in character lengths.
>
> **"masking ratio"**
>
> The average result with a masking ratio of 0.75 is 79.15%. We suppose that the reason our optimal masking ratio is lower than what MAE reported is because text line images have a higher information density compared to the images used in MAE.
>
> **"what are the 1% 10% 100% for?"**
>
> The percentages 1%, 10%, and 100% refer to the proportions of real data used during the fine-tuning stage. Specific details have already been mentioned in the initial submission of the paper. Further detailed descriptions will be provided in the caption of Table 7 to enhance readability.
>
> **"what is the time unit?"**
>
> In Table 10, we evaluate the model's inference speed using "FPS", an abbreviation for "Frames per Second", which denotes the number of images processed per second.

---

### Review · Reviewer_PBQ5 · 2024-02-19

**Summary Of Contributions:**

This paper proposes a masked image-language modeling pretraining scheme to learn a strong encoder-decoder model for the task of text recognition. The proposed pretraining scheme applies masked language modeling to the image encoder over a set of unlabeled real text images, and pretrains the language decoder by transforming a text corpus into images. The method performs well on several Chinese and English benchmarks.

**Audience:**

Yes

**Broader Impact Concerns:**

There are no major issues with the ethical implications of the work. It studies text recognition and is relatively low risk.

**Claims And Evidence:**

Yes

**Requested Changes:**

Would strengthen the work:
- Discussion about model throughput, and why exactly it’s faster than prior approaches

Critical:
- Comparisons against Rust et al. (2023)

**Strengths And Weaknesses:**

In general I like the simplicity of the approach, its strong results, and high inference throughput.


## Strengths:
- The proposed approach is simple and intuitive.
- The model seems to scale well, with results improve substantially from 36M to 100M parameters.
- The ablations are well conducted and informative, and showcase the benefits of the pretraining schemes and architectural design choices.


## Weaknesses:
- What is the reason for the proposed model being so much faster than the other baselines, despite parameter counts being comparable (Table 10)? Could you be more specific in identifying reasons for the speedup?
- The proposed pretraining scheme appears to be the same as the one proposed in [1], but there is no discussion of that paper. Can you provide more explicit comparisons against this work, and highlight the differences?
- Minor typos: “Encode-decoder” -> “Encoder-decoder” (Figure 2 caption), Sec 4.4 (72.8% v.s. 75.6% has incorrect formatting)


**References**

[1] Rust, Phillip, et al. "Language modelling with pixels." ICLR 2023.

---

> ### Author Response · Authors · 2024-02-27
> **Response to reviewer PBQ5**
>
> Thank you for the review and the valuable comments/suggestions! We have attempted to address them  in the responses below.
>
> **Discussion about model throughput**
>
> We measured the computational complexity of the three networks to analyze the difference in inference speed. ABINet has 2.97G Flops, DiG has 15.74G Flops, and MaskOCR has 1.17G Flops, showing a positive correlation with speed. We further investigated why DiG's computational complexity far exceeds that of ABINet and MaskOCR under similar parameters. We found that DiG uses a relatively small patch size (4 * 4), while our approach employs a patch size of 32 * 4. This results in a longer feature sequence input to ViT in DiG, leading to an exponential increase in computational complexity.
>
> **Comparisons against Rust et al.**
>
> We apologize for not comparing our study with [1] and appreciate your correction. The method introduced in [1] is similar to the first stage of our approach, utilizing Masked Image Modeling for pre-training to enhance the encoder's feature extraction. However, a key divergence from [1] lies in our method's incorporation of Masked Image-Language Modeling for additional pre-training of the decoder. This addition aims to enhance the semantic reasoning ability of the decoder when interpreting character sequences.
>
> **Minor typos**
>
> Thanks for pointing that out. We'll carefully go through the paper and fix typos.

---

### Decision · Action_Editor_6Ewh · 2024-04-18

**Recommendation:** Accept as is

**Comment:**

Interesting and novel combination of methods. OCR researchers would likely find it valuable to see the experiments that show how to pretrain the decoder with synthetic images. Lots of interesting ablations.

**Audience:**

Experiments and findings should be useful to the greater text recognition community.

**Claims And Evidence:**

This work describes a novel way to improve encoder-decoder OCR systems by proposing to pretrain the encoder with *real* images, pretraining the decoder with *synthetic* images, and ultimately finetuning the whole thing. The pretraining of separate parts seems to be the key design decision in this case. The experimental evidence suggests that this method is a robust way to get good OCR performance in English and Chinese, as well as having high inference throughput.